# Semi-Supervised Semantic Segmentation via Gentle Teaching Assistant

**Ying Jin**[1]    **Jiaqi Wang**[2][✉]    **Dahua Lin**[1,2]

[1]CUHK-SenseTime Joint Lab, The Chinese University of Hong Kong
[2]Shanghai AI Laboratory
`{jy021,dhlin}@ie.cuhk.edu.hk, wjqdev@gmail.com`

## Abstract

Semi-Supervised Semantic Segmentation aims at training the segmentation model with limited labeled data and a large amount of unlabeled data. To effectively leverage the unlabeled data, pseudo labeling, along with the teacher-student framework, is widely adopted in semi-supervised semantic segmentation. Though proved to be effective, this paradigm suffers from incorrect pseudo labels which inevitably exist and are taken as auxiliary training data. To alleviate the negative impact of incorrect pseudo labels, we delve into the current Semi-Supervised Semantic Segmentation frameworks. We argue that the unlabeled data with pseudo labels can facilitate the learning of representative features in the feature extractor, but it is unreliable to supervise the mask predictor. Motivated by this consideration, we propose a novel framework, **Gentle Teaching Assistant (GTA-Seg)** to disentangle the effects of pseudo labels on feature extractor and mask predictor of the student model. Specifically, in addition to the original teacher-student framework, our method introduces a teaching assistant network which directly learns from pseudo labels generated by the teacher network. The gentle teaching assistant (GTA) is coined gentle since it only transfers the beneficial feature representation knowledge in the feature extractor to the student model in an Exponential Moving Average (EMA) manner, protecting the student model from the negative influences caused by unreliable pseudo labels in the mask predictor. The student model is also supervised by reliable labeled data to train an accurate mask predictor, further facilitating feature representation. Extensive experiment results on benchmark datasets validate that our method shows competitive performance against previous methods. Code is available at `https://github.com/Jin-Ying/GTA-Seg`.

## 1 Introduction

The rapid development in deep learning has brought significant advances to semantic segmentation [29, 5, 52] which is one of the most fundamental tasks in computer vision. Existing methods often heavily rely on numerous pixel-wise annotated data, which is labor-exhausting and expensive. Towards this burden, great interests have been aroused in Semi-Supervised Semantic Segmentation, which attempts to train a semantic segmentation model with limited labeled data and a large amount of unlabeled data.

The key challenge in semi-supervised learning is to effectively leverage the abundant unlabeled data. One widely adopted strategy is pseudo labeling [27]. As shown in Figure 1, the model assigns pseudo labels to unlabeled data based on the model predictions on-the-fly. These data with pseudo labels will be taken as auxiliary supervision during training to boost performance. To further facilitate

---

[✉]Corresponding author.

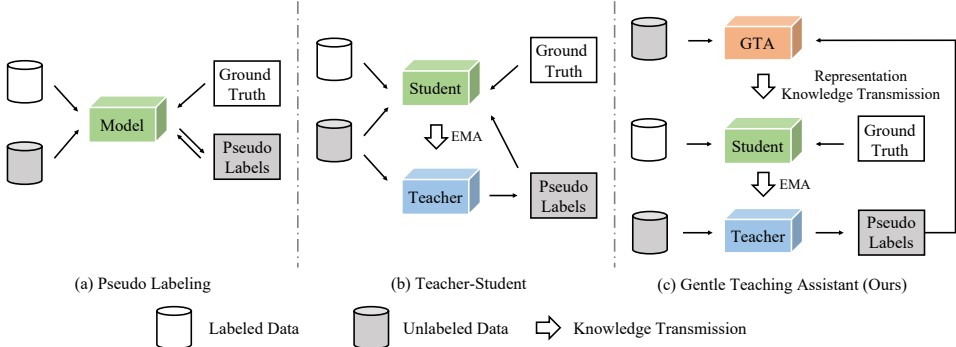

Figure 1: **Comparison with previous frameworks. (a)** The vanilla pseudo labeling framework. The model generates pseudo labels by itself and in turn, learns from them. **(b)** The pseudo labeling with the teacher-student framework. The teacher model is responsible for generating pseudo labels while the student model learns from the pseudo labels and the ground-truth labels simultaneously. Knowledge Transmission is conducted between the two models via Exponential Moving Average (EMA) of all parameters. **(c)** Our method attaches a gentle teaching assistant (GTA) module to the teacher-student framework. Different from the original one in (b), the gentle teaching assistant (GTA) learns from the pseudo labels while the student model only learns from ground-truth labels. We design the representation knowledge transmission between the GTA and student to mitigate the negative influence caused by unreliable pseudo labels.

semi-supervised learning, the teacher-student framework [42, 46, 43] is incorporated. The teacher model, which is the Exponential Moving Average (EMA) of the student model, is responsible for generating smoothly updated pseudo labels. Via jointly supervised by limited data with ground-truth labels and abundant data with pseudo labels, the student model can learn more representative features, leading to significant performance gains.

Although shown to be effective, the pseudo labeling paradigm suffers from unreliable pseudo labels, leading to inaccurate mask predictions. Previous research work alleviates this problem by filtering out predictions that are lower than a threshold of classification scores [3, 39, 50]. However, this mechanism can not perfectly filter out wrong predictions, because some wrong predictions may have high classification scores, named over-confidence or mis-calibration [17] phenomenon. Moreover, a high threshold will heavily reduce the number of generated pseudo labels, limiting the effectiveness of semi-supervised learning.

Towards the aforementioned challenge, it is necessary to propose a new pseudo labeling paradigm that can learn representative features from unlabeled data as well as avoid negative influences caused by unreliable pseudo labels. Delving into the semantic segmentation framework, it is composed of a feature extractor and a mask predictor. Previous works ask the feature extractor and the mask predictor to learn from both ground-truth labels and pseudo labels simultaneously. As a result, the accuracy of the model is harmed by incorrect pseudo labels. To better leverage the unlabeled data with pseudo labels, a viable solution is to let the feature extractor learn feature representation from both ground-truth labels and pseudo labels, while the mask predictor only learns from ground-truth labels to predict accurate segmentation results.

Accordingly, we propose a novel framework, Semi-Supervised Semantic Segmentation via **Gentle Teaching Assitant (GTA-Seg)**, which attaches an additional gentle teaching assistant (GTA) module to the original teacher-student framework. Figure 1 compares our method with previous frameworks. In our method, the teacher model generates pseudo labels for unlabeled data and the gentle teaching assistant (GTA) learns from these unlabeled data. Only knowledge of the feature extractor in the gentle teacher assistant (GTA) is conveyed to the feature extractor of the student model via Exponential Moving Average (EMA). We coin this process as **representation knowledge transmission**. Meanwhile, the student model also learns from the reliable ground-truth labels to optimize both the feature extractor and mask predictor. The gentle teaching assistant (GTA) is called gentle since it not only transfers the beneficial feature representation knowledge to the student model, but also protects the student model from the negative influences caused by unreliable pseudo labels in

the mask predictor. Furthermore, a re-weighting mechanism is further adopted for pseudo labels to suppress unreliable pixels.

Extensive experiments have validated that our method shows competitive performance on mainstream benchmarks, proving that it can make better utilization of unlabeled data. In addition, we can observe from the visualization results that our method boasts clearer contour and more accurate classification for objects, which indicates better segmentation performance.

## 2  Related Work

**Semantic Segmentation**  Semantic Segmentation, aiming at predicting the label of each pixel in the image, is one of the most fundamental tasks in computer vision. In order to obtain the dense predictions, FCN [29] replaces the original fully-connected layer in the classification model with convolution layers. The famous encoder-decoder structure is borrowed to further refine the pixel-level outputs [34, 2]. Meanwhile, intensive efforts have been made to design network components that are suitable for semantic segmentation. Among them, dilated convolution [48] is proposed to enhance receptive fields, global and pyramid pooling [28, 5, 52] are shown to be effective in modeling context information, and various attention modules [51, 53, 14, 21, 41] are adopted to capture the pixel relations in images. These works mark milestones in this important computer vision task, but they pay rare attention to the data-scarce scenarios.

**Semi-Supervised Learning**  Mainstream methods in Semi-Supervised Learning [55] (SSL) fall into two lines of work, self-training [16, 27] and consistency regularization [26, 38, 33, 45, 42]. The core spirit of self-training is to utilize the model predictions to learn from unlabeled data. Pseudo Labeling [27], which converts model predictions on unlabeled data to one-hot labels, is a widely-used technique [3, 39, 50] in semi-supervised learning. Another variant of self-training, entropy minimization [37], is also proved to be effective both theoretically [44] and empirically [16]. Consistency Regularization [38, 45] forces the model to obtain consistent predictions when perturbations are imposed on the unlabeled data. Some recent works unveil that self-training and consistency regularization can cooperate harmoniously. MixMatch [3] is a pioneering holistic method and boasts remarkable performance. On the basis of MixMatch, Fixmatch [39] further simplify the learning process while FlexMatch [50] introduces a class-wise confidence threshold to boost model performance.

**Semi-Supervised Semantic Segmentation**  Semi-Supervised Semantic Segmentation aims at pixel-level classification. Borrowing the spirit of Semi-Supervised Learning, self-training and consistency regularization gives birth to various methods. One line of work [56, 7, 20, 43] applies pseudo labeling in self-training to acquire auxiliary supervision, while methods based on consistency [32] pursue stable outputs at both feature [25, 54] and prediction level [36]. Apart from them, Generative Adversarial Networks (GANs) [15] or adversarial learning are often leveraged to provide additional supervision in relatively early methods [40, 22, 31, 24]. Various recent methods tackles this problem from other perspectives, such as self-correcting networks [23] and contrastive learning [1]. Among them, some works [49] unveil another interesting phenomenon that the most fundamental training paradigm, equipped with strong data augmentations, can serve as a simple yet effective baseline. In this paper, we shed light on semi-supervised semantic segmentation based on pseudo labeling and strives to alleviate the negative influence caused by noisy pseudo labels.

## 3  Method

### 3.1  Preliminaries

**Semi-Supervised Semantic Segmentation**  In Semi-Supervised Semantic Segmentation, we train a model with limited labeled data $D_l = \{x_i^l, y_i^l\}_{i=1}^{N^l}$ and a large amount of unlabeled data $D_u = \{x_i^u\}_{i=1}^{N^u}$, where $N^u$ is often much larger than $N^l$. The semantic segmentation network is composed of the feature extractor $f$ and the mask predictor $g$. The key challenge of Semi-Supervised Semantic Segmentation is to make good use of the numerous unlabeled data. And one common solution is pseudo labeling [27, 47].

**Pseudo Labeling** Pseudo Labeling is a widely adopted technique for semi-supervised segmentation, which assigns pseudo labels to unlabeled data according to model predictions on-the-fly. Assuming there are $\mathcal{K}$ categories, considering the $j^{th}$ pixel on the $i^{th}$ image, the model prediction $p_{ij}^u$ and the corresponding confidence $c_{ij}^u$ will be

$$p_{ij}^u = g(f(x_{ij}^u)), \; c_{ij}^u = \max_k p_{ij}^u, \; \text{with } k \in \mathcal{K}, \tag{1}$$

where $k$ denotes the $k-th$ category, larger $c_{ij}^u$ indicates that the model is more certain on this pixel, which is consequently, more suitable for generating pseudo labels. Specifically, we often keep the pixels whose confidence value is greater than one threshold, and generate pseudo labels as

$$\hat{y}_{ij}^u = \begin{cases} \arg\max_k p_{ij}^u, & c_{ij}^u > \gamma_t \\ \text{ignore}, & \text{otherwise} \end{cases}, \tag{2}$$

where $\gamma_t$ is the confidence threshold at the $t$ iteration. We note that $\gamma_t$ can be a constant or a varied value during training. The $j^{th}$ pixel on $i^{th}$ image with a confidence value larger than $\gamma_t$ will be assigned with pseudo label $\hat{y}_{ij}^u$. The unlabeled data that are assigned with pseudo labels will be taken as auxiliary training data, while the other unlabeled data will be ignored.

**Teacher-Student Framework** Teacher-Student [9, 42, 43] framework is a currently widely applied paradigm in Semi-Supervised Segmentation, which consists of one teacher model and one student model. The teacher model is responsible for generating pseudo labels while the student model learns from both the ground-truth labels and pseudo labels. Therefore, the loss for the student model is

$$L = L_l + \mu L_u, L_u = \sum_i \sum_j L_{ce}(p_{ij}^u, \hat{y}_{ij}^u) \tag{3}$$

In Semi-Supervised Semantic Segmentation, $L_l$ and $L_u$ are the cross-entropy loss on labeled data and unlabeled data with pseudo labels, respectively [43], and $\mu$ is a loss weight to adjust the trade-off between them. The optimization of the student model can be formulated as

$$\theta^{student} := \theta^{student} - \lambda \frac{\partial L}{\partial \theta^{student}}, \tag{4}$$

where $\lambda$ denotes the learning rate. In the Teacher-Student framework, after the parameters of the student model are updated, the parameters of the teacher model will be updated by the student parameters in an Exponential Moving Average (EMA) manner.

$$\theta^{teacher}(t) = \alpha\theta^{teacher}(t-1) + (1-\alpha)\theta^{student}(t), \tag{5}$$

where $\theta^{teacher}(t)$ and $\theta^{student}(t)$ denote the parameters of the teacher and student model at $t$-th iteration, respectively. $\alpha$ is a hyper-parameter in EMA, where $\alpha \in [0,1]$.

### 3.2 Gentle Teaching Assistant

In this section, we will introduce our Gentle Teaching Assistant framework for semi-supervised semantic segmentation (GTA-Seg), as shown in Figure 2, which consists of the following three steps.

**Step 1: Pseudo Label Generation and Re-weighting.** Similar to previous work [43], the teacher model is responsible for generating pseudo labels. A confidence threshold is also adopted to filter out the pseudo labels with low confidence. For the kept pixels, instead of treating all of them equally, we propose a re-weighting mechanism according to the confidence of each pixel as follows,

$$w_{ij}^u = \frac{(c_{ij}^u + \tau) \cdot \mathbb{1}(c_{ij}^u > \gamma_t)}{\sum_i \sum_j (c_{ij}^u + \tau) \cdot \mathbb{1}(c_{ij}^u > \gamma_t)} \cdot \sum_i \sum_j \mathbb{1}(c_{ij}^u > \gamma_t). \tag{6}$$

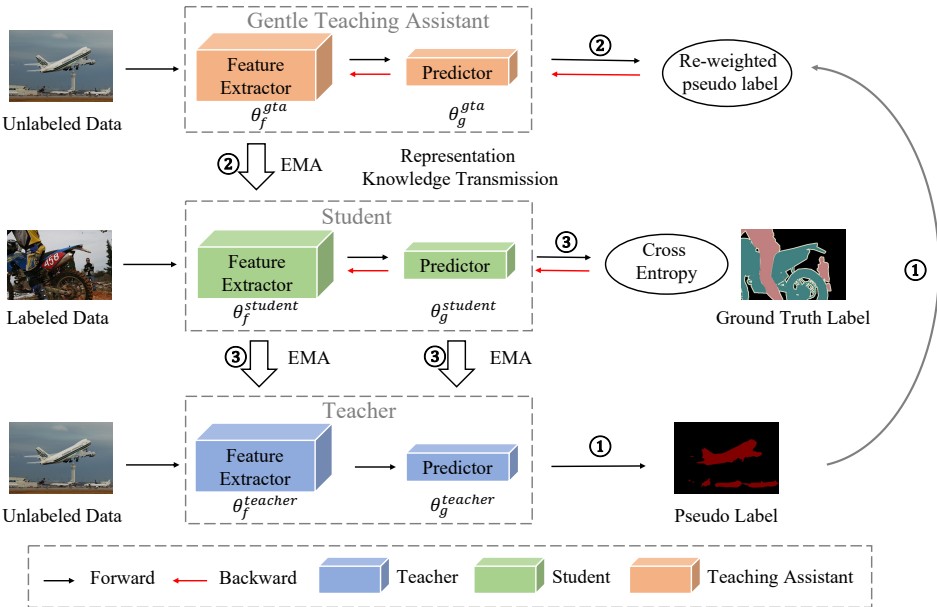

Figure 2: **Method Overview.** Our Gentle Teaching Assistant (GTA) framework can be divided into three steps. **Step 1**: The teacher model generates pseudo labels and then the gentle teaching assistant can learn from them. One re-weighting strategy is incorporated to assign importance weights to the generated pseudo labels. **Step 2**: The gentle teaching assistant model learns from the pseudo labels and performs representation knowledge transmission, which only conveys the learned knowledge in the feature extractor to the student model via Exponential Moving Average (EMA). **Step 3**: After absorbing the knowledge from our gentle teaching assistant, the student model learns from ground-truth labels and optimizes all parameters. Finally, the parameters of the teacher model will also be updated according to the student model via EMA at the end of each training iteration.

In our re-weighting strategy, the pixel with higher confidence will be highlighted while the other will be suppressed. As a result, the negative influence caused by unreliable pseudo labels can be further alleviated. We adopt Laplace Smoothing [30] to avoid over penalization where $\tau$ is a predefined coefficient. With this re-weighting mechanism, the unsupervised loss on unlabeled data becomes

$$L_u = \sum_i \sum_j w_{ij}^u L_{ce}(p_{ij}^u, \hat{y}_{ij}^u). \tag{7}$$

**Step 2: Representation Knowledge Transmission via Gentle Teaching Assistant (GTA).** Gentle Teaching Assistant (GTA) plays a crucial role in our framework. Previous works force the student model to learn from both labeled and unlabeled data simultaneously. We argue that it is dangerous to treat ground-truth labels and pseudo labels equally since the incorrect pseudo labels will mislead the mask prediction. Therefore, we want to disentangle the effects of pseudo labels on feature extractor and mask predictor of the student model. Concretely, our solution is to introduce one additional gentle teaching assistant, which learns from the unlabeled data and only transfers the beneficial feature representation knowledge to the student model, protecting the student model from the negative influences caused by unreliable pseudo labels.

After optimized on unlabeled data with pseudo labels as in Eq. 8, the gentle teaching assistant model is required to convey the learned representation knowledge in feature extractor to the student model via Exponential Moving Average (EMA) as in Eq. 9,

$$\theta^{gta} := \theta^{gta} - \lambda \frac{\partial L_u}{\partial \theta^{gta}}. \tag{8}$$

$$\theta_f^{student}(t) = \alpha \theta_f^{student}(t-1) + (1-\alpha)\theta_f^{gta}(t), \tag{9}$$

---

**Algorithm 1:** Gentle Teaching Assistant for Semi-Supervised Semantic Segmentation (GTA-Seg).

---

**Input** : Labeled data $D_l = \{x_i^l, y_i^l\}_{i=1}^{N^l}$, unlabeled data $D_u = \{x_i^u\}_{i=1}^{N^u}$, batch size $B$
**Output** : Teacher Model

**1** Initialization;
**2 for** *minibatch* $\{x^l, y^l\}, \{x^u\} \in \{D_l, D_u\}$ **do**
**3**     Step 1:
**4**        Teacher model generates pseudo labels on $x^u$ samples by Eq. (2);
**5**        Re-weight pseudo labels by Eq. (6) and compute unsupervised loss $L_u$ by Eq. (7);
**6**     Step 2:
**7**        Update Gentle Teaching Assistant (GTA) by unlabeled data Eq. (8);
**8**        Representation knowledge Transmission from GTA to student by Eq. (9);
**9**     Step 3:
**10**        Compute supervised loss on $\{x^l, y^l\}$ by Eq. (10) ;
**11**        Update student model by labeled data via Eq. (11) ;
**12**        Update teacher model by Eq. (12) ;
**13 end**

---

where $\theta^{gta}(t)$ is the parameters of the gentle teaching assistant model at $t$-th iteration, $\theta^{student}(t)$ is the parameters of the student model at $t$-th iteration, and $\theta_f$ denotes the parameters of the feature extractor. Through our representation knowledge transmission, the unlabeled data is leveraged to facilitate feature representation of the student model, but it will not train the mask predictor.

**Step 3: Optimize student model with ground truth labels and update teacher model.** With the gentle teaching assistant module, the student model in our framework is only required to learn from the labeled data,

$$L_l = \sum_i \sum_j L_{ce}(p_{ij}^l, y_{ij}^l), \tag{10}$$

$$\theta^{student} := \theta^{student} - \lambda \frac{\partial L_l}{\partial \theta^{student}}. \tag{11}$$

Here, the whole model, including the feature extractor as well as the mask predictor, is updated according to the supervised loss computed by the ground-truth labels of labeled data.

Then the teacher model is updated by taking the EMA of the student model according to the traditional paradigm in the teacher-student framework.

$$\theta_f^{teacher}(t) = \alpha \theta_f^{teacher}(t-1) + (1-\alpha)\theta_f^{student}(t),$$
$$\theta_g^{teacher}(t) = \alpha \theta_g^{teacher}(t-1) + (1-\alpha)\theta_g^{student}(t). \tag{12}$$

Finally, the teacher model, which absorbs the knowledge of both labeled and unlabeled data from the student model, will be taken as the final model for inference.

## 4 Experiment

### 4.1 Datasets

We evaluate our method on **1) PASCAL VOC 2012** [11]: a widely-used benchmark dataset for semantic segmentation, with 1464 images for training and 1449 images for validation. Some researches [7, 47] augment the training set by incorporating the 9118 coarsely annotated images in SBD [18] to the original training set, obtaining 10582 labeled training images, which is called the augmented training set. In our experiments, we consider both the original training set and the augmented training set, taking 92, 183, 366, 732, and 1464 images from the 1464 labeled images in the original training set, and 662, 1323 and 2645 images from the 10582 labeled training images in

Table 1: Results on PASCAL VOC 2012, original training set. We have 1464 labeled images in total and sample different proportions of them as labeled training samples. SupOnly means training the model merely on the labeled data, with all the other unlabeled data abandoned. All the other images in the training set (including images in the augmented training set) are used as unlabeled data. We use ResNet-101 as the backbone and DeepLabv3+ as the decoder.

| Method | 92 | 183 | 366 | 732 | 1464 |
|---|---|---|---|---|---|
| SupOnly | 45.77 | 54.92 | 65.88 | 71.69 | 72.50 |
| MT [42] | 51.72 | 58.93 | 63.86 | 69.51 | 70.96 |
| CutMix [13] | 52.16 | 63.47 | 69.46 | 73.73 | 76.54 |
| PseudoSeg [56] | 57.60 | 65.50 | 69.14 | 72.41 | 73.23 |
| PC2Seg [54] | 57.00 | 66.28 | 69.78 | 73.05 | 74.15 |
| ST++ [47] | 65.23 | 71.01 | 74.59 | 77.33 | 79.12 |
| U2PL [43] | 67.98 | 69.15 | 73.66 | 76.16 | 79.49 |
| GTA-Seg (Ours) | **70.02** $\pm$ 0.53 | **73.16** $\pm$ 0.45 | **75.57** $\pm$ 0.48 | **78.37** $\pm$ 0.33 | **80.47** $\pm$ 0.35 |

Table 2: Results on PASCAL VOC 2012, augmented training set. We have 10582 labeled images in total and sample different proportions of them as labeled training samples. All the other images in the training set are used as unlabeled data. The notations and network architecture are the same as in Table 1.

| Method | 662 | 1323 | 2645 | 5291 |
|---|---|---|---|---|
| MT [42] | 70.51 | 71.53 | 73.02 | 76.58 |
| CutMix [13] | 71.66 | 75.51 | 77.33 | 78.21 |
| CCT [36] | 71.86 | 73.68 | 76.51 | 77.40 |
| GCT [24] | 70.90 | 73.29 | 76.66 | 77.98 |
| CPS [7] | 74.48 | 76.44 | 77.68 | 78.64 |
| AEL [20] | 77.20 | 77.57 | 78.06 | 80.29 |
| GTA-Seg (Ours) | **77.82** $\pm$ 0.31 | **80.47** $\pm$ 0.28 | **80.57** $\pm$ 0.33 | **81.01** $\pm$ 0.24 |

Table 3: Results on Cityscapes dataset. We have 2975 labeled images in total and sample different proportions of them as labeled training samples. The notations and network architecture are the same as in Table 1. * means that we reimplement the method with ResNet-101 backbone for a fair comparison.

| Method | 100 | 186 | 372 | 744 |
|---|---|---|---|---|
| DMT [12] | 54.82 | - | 63.01 | - |
| CutMix [13] | 55.73 | 60.06 | 65.82 | 68.33 |
| ClassMix [35] | - | 59.98 | 61.41 | 63.58 |
| Pseudo-Seg [56] | 60.97 | 65.75 | 69.77 | 72.42 |
| DCC* [25] | 61.15 | 67.74 | 70.45 | 73.89 |
| GTA-Seg (Ours) | **62.95** $\pm$ 0.32 | **69.38** $\pm$ 0.24 | **72.02** $\pm$ 0.32 | **76.08** $\pm$ 0.25 |

the augmented training set. **2) Cityscapes** [8], a urban scene dataset with 2975 images for training and 500 images for validation. We sample 100, 186, 372, 744 images from the 2975 labeled images in the training set. We take the split in [56] and report all the performances in a fair comparison.

## 4.2 Implementation Details

We take ResNet-101 [19] pre-trained on ImageNet [10] as the network backbone and DeepLabv3+ [6] as the decoder. The segmentation head maps the 512-dim features into pixel-wise class predictions.

We take SGD as the optimizer, with an initial learning rate of 0.001 and a weight decay of 0.0001 for PASCAL VOC. The learning rate of the decoder is 10 times of the network backbone. On Cityscapes, the initial learning rate is 0.01 and the weight decay is 0.0005. Poly scheduling is applied to the learning rate with $lr = lr_{init} \cdot (1 - \frac{t}{T})^{0.9}$, where $lr_{init}$ is the initial learning rate, $t$ is the current iteration and $T$ is the total iteration. We take 4 GPUs to train the model on PASCAL VOC, and 8 GPUs on Cityscapes. We set the trade-off between the loss of labeled and unlabeled data $\mu = 1.0$, the hyper-parameter $\tau = 1.0$ in our re-weighting strategy and the EMA hyper-parameter $\alpha = 0.99$ in all of our experiments. At the beginning of training, we train all three components (the gentle teaching assistant, the student and the teacher) on labeled data for one epoch as a warm-up following conventions [42], which enables a fair comparison with previous methods. Then we continue to train the model with our method. For pseudo labels, we abandon the 20% data with lower confidence. We run each experiment 3 times with random seed = 0, 1, 2 and report the average results. Following previous works, input images are center cropped in PASCAL VOC during evaluation, while on Cityscapes, sliding window evaluation is adopted. The mean of Intersection over Union (mIoU) measured on the validation set serves as the evaluation metric.

Table 4: Ablation Study on the components in our method, on the original training set of PASCAL VOC 2012, with 183 labeled samples.

| Teacher-Student | Gentle Teaching Assistant | Re-weighted | mIoU |
|:---:|:---:|:---:|:---|
| ✗ | ✗ | ✗ | 54.92 |
| ✓ | ✗ | ✗ | 58.93 |
| ✓ | ✓ | ✗ | 72.10 |
| ✓ | ✓ | ✓ | 73.16 |

Table 5: Comparison of knowledge transmission mechanisms. The experiment settings follow Table 4.

| Method | mIoU |
|:---|:---|
| SupOnly | 54.92 |
| Original EMA (all parameters) | 64.07 |
| Unbiased ST [4] | 65.92 |
| EMA (Encoder) (Ours) | 72.10 |

## 4.3 Experimental Results

**PASCAL VOC 2012**   We first evaluate our method on the original training set of PASCAL VOC 2012. The results in Table 1 validate that our method surpasses previous methods by a large margin. Specifically, our method improves the supervised-only (SupOnly) model by 24.25, 18.24, 9.69, 6.68, 7.97 in mIoU when 0.9%, 1.7%, 3.4%, 7.0%, 13.9% of the data is labeled, respectively. When compared to the readily strong semi-supervised semantic segmentation method, our method still surpasses it by 13.02, 6.88, 5.79, 5.32, 6.32 respectively. We note that in the original training set, the ratio of labeled data is relatively low (0.9% to 13.9%). Therefore, the results verify that our method is effective in utilizing unlabeled data in semi-supervised semantic segmentation.

We further compare our method with previous methods on the augmented training set of PASCAL VOC 2012, where the annotations are relatively low in quality since some of labeled images come from SBD [18] dataset with coarse annotations. We can observe from Table 2, our method consistently outperforms the previous methods in a fair comparsion.

**Cityscapes**   For Cityscapes, as shown in Table 3, our method still shows competitive performance among previous methods, improving the existing state-of-the-art method by 1.80, 1.64, 1.57, 2.19 in mIoU when 3.3%, 6.25%, 12.5%, 25.0% of the data is labeled.

## 4.4 Analyses

**Component Analysis**   We analyze the effectiveness of different components in our method, *i.e.*, the original teacher-student framework, gentle teaching assistant and re-weighted pseudo labeling as in Table 4. According to the results in Table 4, the carefully designed gentle teaching assistant mechanism (the third row) helps our method outperform the previous methods, pushing the performance about 13.1 higher than the original teacher-student model (the second row). Further, the re-weighted pseudo labeling brings about 1.1 performance improvements. With all of these components, our method outperforms the teacher-student model by over 14.0 and SupOnly by over 18.0 in mIoU.

**Gentle Teaching Assistant**   As mentioned in Table 4, our proposed gentle teaching assistant framework brings about remarkable performance gains. Inspired by this, we delve deeper into the gentle teaching assistant model in our framework. We first consider the representation knowledge transmission mechanism. In Table 5, we compare our mechanism with other methods such as the original EMA [42] that updates all of the parameters via EMA and Unbiased ST [4] that introduces an additional agent to convey representation knowledge. We can observe that all these mechanisms boost SupOnly remarkably, while our mechanism is superior to other methods.

We next pay attention to the three models in our framework, *i.e.* gentle teaching assistant model, student model, and teacher model. Table 6 reports the evaluation performance of them. All of them show relatively competitive performance. For the teacher assistant model, it is inferior to the student model. This is reasonable since it is only trained on pseudo labels, while the student model inherits the representation knowledge of unlabeled data from the gentle teaching assistant as well as trained on labeled data. In addition, the teacher model performs best, which agrees with previous works [42].

**Method Design**   In our method, we train GTA with pseudo labels and the student model with labeled data. It is interesting to explore the model performance of other designs. Table 9 shows that 1) training the student model with pseudo labels will cause significant performance drop, which is consistent with our statement that the student model shall not learn from the pseudo labels directly. 2) Incorporating labeled data in training GTA is not beneficial to model performance. We conjecture

Table 6: Results of the three models on the original PASCAL VOC 2012. The experiment settings follow Table 4.

| Method | mIoU |
|---|---|
| Gentle Teaching Assistant | 70.10 |
| Student Model | 72.71 |
| Teacher Model | 73.16 |

Table 7: Ablation study on our method design on the original PASCAL VOC 2012. The experiment settings follow Table 4.

| Gentle Teaching Assistant | Student | mIoU |
|---|---|---|
| Labeled Data | Pseudo Labels | 66.71 |
| Labeled Data + Pseudo Labels | Labeled Data | 72.28 |
| Pseudo Labels | Labeled Data | 73.16 |

Table 8: Performance under different EMA hyper-parameters and warmup epochs. The experiment settings follow Table 4.

| $\alpha$ | mIoU | warmup | mIoU |
|---|---|---|---|
| 0.99 (Reported) | 73.16 | 1 (Reported) | 73.16 |
| 0.999 | 73.44 | 2 | 73.58 |
| 0.9999 | 73.57 | 3 | 73.39 |

Table 9: Ablation study on our re-weighting strategy for pseudo labeling on the original PASCAL VOC 2012. The experiment settings follow Table 4.

| Confidence-based Re-weighting | Laplace Smoothing | mIoU |
|---|---|---|
| ✗ | ✗ | 72.10 |
| ✓ | ✗ | 70.67 |
| ✓ | ✓ | 73.16 |

that when we transmit the knowledge of labeled data from GTA to the student model, as well as supervise the student model with labeled data, the limited labels overwhelm the updating of the student model, which possibly leads to overfitting and harms the student model's performance. Then since the teacher model is purely updated by the student model via EMA, the performance of the teacher model is also harmed. Considering the ultimate goal is a higher performance of the teacher model, we choose to train GTA with pseudo labels alone.

**Re-weighting strategy** In our method, we design the re-weighting strategy for pseudo labels as Eq. 6, which contains 1) confidence-based re-weighting, 2) Laplace Smoothing. Here we conduct further ablation study on our design. Table 9 shows that though effective in other tasks such as semi-supervised object detection [46], in our framework, adopting confidence-base re-weighting is harmful, dropping the performance from 72.10 to 70.67. On the contrary, our strategy, with the help of Laplace Smoothing [30] which alleviates over-penalization, pushes the readily strong performance to a higher level.

**Hyper-parameter sensitivity** We evaluate the performance of our method under different EMA hyper-parameters and various warmup epochs. Results in Table 8 demonstrates that our method performs steadily under different hyper-parameters. In addition, the performance can still be slightly enhanced if the hyper-parameters are tuned carefully.

**Visualization** Besides quantitative results, we present the visualization results to further analyze our method. We note that the model is trained on as few as 183 labeled samples and about 10400 unlabeled samples. As shown in Figure 3, facing such limited labeled data, training the model merely in the supervised manner (SupOnly) appears to be vulnerable. Under some circumstances, the model is even ignorant of the given images (the third and the fourth row). While methods that utilize unlabeled data (teacher-student model and our method), show stronger performance. Further, compared with the original teacher-student model, our method shows a stronger ability in determining a clear contour of objects (the first row) and recognizing the corresponding categories (the second row). Our method is also superior to previous methods in distinguishing objects from the background (the third and fourth row).

In addition, we present more visualization results about our designed re-weighting strategy. We can observe from Figure 4 that incorporating the re-weighting strategy into our method leads to better performance on contour or ambiguous regions.

**Limitations** One limitation of our method is that it brings about more training costs since it incorporates an extra gentle teaching assistant model. Fortunately, the inference efficiency is not influenced since only the teacher model is taken for inference. On the other hand, our method only attempts at making better use of the unlabeled data, but little attention has been paid to the labeled

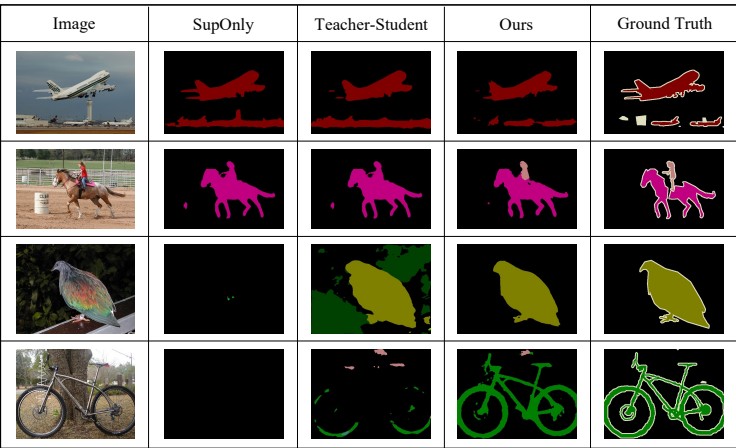

Figure 3: **Visualization Results** on PASCAL VOC 2012, with the original training set. We train the model with 183 labeled data, the other settings are the same as Table 1. From left to right, we show the raw images, results on SupOnly (the model trained merely on labeled data), Teacher-Student model, and our method, as well as the ground truth respectively.

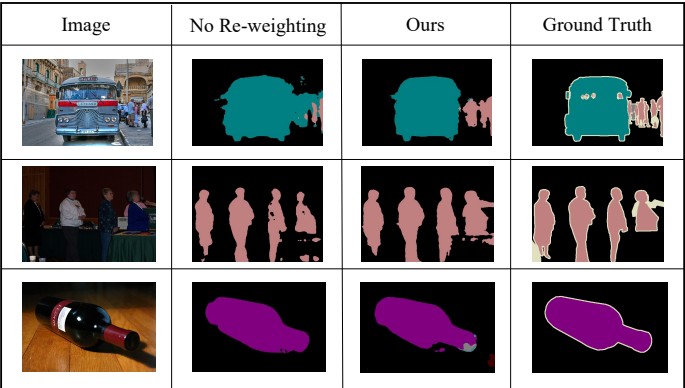

Figure 4: **Visualization Results** on PASCAL VOC 2012, with the original training set. We train the model with 183 labeled data, the other settings are the same as Table 1. From left to right, we show the raw images, our method without our re-weighting mechanism, and our method with re-weighting, as well as the ground truth respectively.

data. We consider it promising to conduct research on how to better leverage the labeled data in semi-supervised semantic segmentation.

## 5 Conclusion

In this paper, we propose a novel framework, Gentle Teaching Assistant, for semi-supervised semantic segmentation (GTA-Seg). Concretely, we attach an additional teaching assistant module to disentangle the effects of pseudo labels on the feature extractor and the mask predictor. GTA learns representation knowledge from unlabeled data and conveys it to the student model via our carefully designed representation knowledge transmission. Through this framework, the model optimizes representation with unlabeled data, as well as prevents it from overfitting on limited labeled data. A confidence-based pseudo label re-weighting mechanism is applied to further boost the performance. Extensive experiment results prove the effectiveness of our method.

**Acknowledgements.** This work is supported by GRF 14205719, TRS T41-603/20-R, Centre for Perceptual and Interactive Intelligence, CUHK Interdisciplinary AI Research Institute, and Shanghai AI Laboratory.

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
