# OpenReview forum: "Semi-Supervised Semantic Segmentation via Gentle Teaching Assistant"
_NeurIPS.cc/2022/Conference — NeurIPS 2022 Accept_

### Official Review · Reviewer_rFZT · 2022-07-08

**Rating:** 6
**Confidence:** 5
**Soundness:** 3 good
**Presentation:** 3 good
**Contribution:** 3 good

**Summary:**

The paper proposes a semi-supervised method for semantic segmentation via a teacher-student framework. Unlike the standard approach that the teacher model generates pseudo-labels for unlabeled data, while the student model takes both labeled and pseudo-labeled data as training signals, the authors observe that pseudo-labels can be noisy and directly using them to train the student model may not be optimal. Therefore, the paper proposes to use another assistant model that takes pseudo-labels and only transfers feature representations to the student model via EMA. In addition, the paper also adopts a weighted pseudo-labeling scheme to train the assistant model. Experiments are conducted on PASCAL VOC and Cityscapes to demonstrate the usefulness of introducing the assistant model.

**Questions:**

Please see the above comments in Weakness.

**Limitations:**

Limitations are briefly discussed at the end of the paper. However, the statements may not be correct. The authors state that the improvement in Table 2 is smaller than the one in Table 1, but their compared methods are not the same though. For example, the improvement over CutMix in Table 1 and 2 are actually similar. Also, when the labeled data is around 700 or 1400, the final results in Table 1 and 2 are similar. Although the labeled data can be noisy in the augmented set as the authors mentioned, I would suggest the authors to discuss the limitation from other perspectives.

**Strengths And Weaknesses:**

**Strength**
- The paper is written well and is easy to follow
- The proposed method with the assistant model is simple and effective, which can reduce the issue of noisy pseudo-labels
- The proposed feature transmission from the assistant model to the student model can be a generally useful tool for other tasks
- Experiments show good performance improvements on two datasets

**Weakness**

Some hyperparameters and technical details are not explained very well
- What is the \tau value in Eq. (6)?
- How do the authors select the threshold for pseudo-labeling, i.e., is \gamma a constant or varied and how?

Experimental results:
- Table 4-7 are conducted on a smaller labeled set (183 images). However, I wonder how Table 4-7 would look like when there are more labeled data, e.g., >1000/2000 labeled images, which can be a more practical situation in real applications.
- In Table 6, the teacher model performs better than the student model. It's interesting to know whether it is the same case for all experimental settings. If so, what is the intuition behind it? The authors may have more discussions on this.

Missing references:
- Adversarial Learning for Semi-Supervised Semantic Segmentation, BMVC'18
- Semi-supervised semantic segmentation with high-and low-level consistency, PAMI'19
- Semi-Supervised Semantic Image Segmentation With Self-Correcting Networks, CVPR'20
- Semi-Supervised Semantic Segmentation With Pixel-Level Contrastive Learning From a Class-Wise Memory Bank, ICCV'21

---

> ### Author Response · Authors · 2022-08-02
> **Response to Reviewer rFZT**
>
> Sincere gratitude to you for the recognition of our idea and detailed suggestions. We hope our answers can address all of your concerns. We are looking forward to having further discussions with you.
>
> __Q1: What is the \tau value in Eq. (6)?__
>
>
> The hyper-parameter $\tau$ is 1.0 in all of the experiments.
>
>
> __Q2: How do the authors select the threshold for pseudo-labeling, i.e., is \gamma a constant or varied and how?__
>
>
> We select the top 80% of the data, in other words, we abandon the 20% data with lower confidence than others. $\gamma$ is decided by this criterion so it varies during the training procedure.
>
>
>   __Q3: Table 4-7 are conducted on a smaller labeled set (183 images). However, I wonder how Table 4-7 would look like when there are more labeled data, e.g., >1000/2000 labeled images ...__
>
>
> We have conducted the ablation experiments with 1464 labeled images. The results are shown below.
>
> The ablation study on the components of our method.
>
> | Teacher-Student | Gentle Teaching Assistant | Re-weighted | mIoU |
> |:---:|:---:|:---:|---|
> | &#10004; | x | x | 70.96 |
> | &#10004; | &#10004; | x | 78.95 |
> | &#10004; | &#10004; | &#10004; | __80.47__ |
>
> Comparison of knowledge transmission mechanisms.
>
> | Method | mIoU |
> |-----|----|
> | SupOnly | 72.50 |
> | Original EMA | 74.10 |
> | Unbiased ST | 74.16 |
> | Ours | __78.95__ |
>
> Results of three models.
>
> | Method | mIoU |
> |-----|----|
> | Gentle Teaching Assistant | 76.89 |
> | Student Model | 78.52 |
> | Teacher Model | __78.95__ |
>
> Ablation study of our re-weighting strategy.
>
> | Confidence-based Re-weighting | Laplace Smoothing| mIoU |
> |:---:|:---:|---|
> | x | x | 78.95|
> | &#10004; | x | 79.74|
> | &#10004; | &#10004; | __80.47__ |
>
>
>  __Q4: In Table 6, the teacher model performs better than the student model. It's interesting to know whether it is the same case for all experimental settings. If so, what is the intuition behind it?__
>
>
> Actually, it is the same case in all of our experiments, the teacher model shows slightly stronger performance than the student model. We conjecture that with EMA parameter updates, the teacher model can be optimized more smoothly than the student model and can have a better performance. Fortunately, the stronger teacher model, which generates pseudo labels, can push the performance of our GTA and student model to a higher level by the fly-wheel effect.
>
>
> __Q5: Missing references.__
>
>
> We have added these related works in our revision.
>
>
> __Q6: Questions about limitation.__
>
> Thank you for pointing out this. We express sincere apology that our statement is somewhat confusing. We have analyzed the limitation of our method from other perspectives in our revision.

---

### Official Review · Reviewer_odPv · 2022-07-11

**Rating:** 6
**Confidence:** 3
**Soundness:** 3 good
**Presentation:** 3 good
**Contribution:** 3 good

**Summary:**

This paper addresses the semi-supervised semantic segmentation problem. A teaching assistant model is incorporated into the teacher-student mutual learning framework, which disentangles the effects of pseudo labels on feature extractor and mask predictor and protects the student model from the negative influences caused by unreliable pseudo labels in the mask predictor. Results on benchmark datasets show competitive performance against previous methods.

**Questions:**

The authors seemed not taken the liberty to establish statistical significance of the experimental results. Did the results the average of several experiments with different random seeds? They do not but promise to release the code upon the paper accepted.

**Limitations:**

Yes, the authors have briefly discussed the limitations of the proposed method in the end of the experimental section.

**Strengths And Weaknesses:**

The proposed method is simple yet effective. The idea of gentle teaching assistant is novel. Although decomposing the segmentation task into feature extraction and mask prediction is not new, disentangling the effects of pseudo labels for the tasks in new and interesting. The paper is well written and easy to read.

---

> ### Author Response · Authors · 2022-08-02
> **Response to Reviewer odPv**
>
> Sincere gratitude to you. We hope our answers can address all of your concerns. We are looking forward to having further discussions with you.
>
> __Q1: The authors seemed not taken the liberty to establish statistical significance of the experimental results. Did the results the average of several experiments with different random seeds?__
>
> We run experiments 3 times and report the average score (Appendix A.1), with random seed = 0, 1, 2 respectively. In addition, we have reported the standard deviation (std) in our experiments in the revision, please refer to Table 1-3 in the revised paper for details.
>
> __Q2: They do not but promise to release the code upon the paper accepted.__
>
> We promise to release all the code and models of our method after the paper is accepted.

---

### Official Review · Reviewer_sFdH · 2022-07-11

**Rating:** 4
**Confidence:** 4
**Soundness:** 2 fair
**Presentation:** 3 good
**Contribution:** 2 fair

**Summary:**

The paper tackles the problem of semi-supervised learning (limited labeled data and a higher amount of unlabeled data) in the context of semantic segmentation. The authors adopt an auxiliary teaching assistant as an extension of the classical teacher-student paradigm. The proposed GTA module is trained solely on the unlabeled data using predictions from the original teacher. GTA facilitates feature representation learning that is further passed down to the student network without impacting the mask predictor from incorrect pseudo labels (trained supervised). Results are showcased on popular benchmarks, such as PASCAL VOC 2012 and Cityscapes.


**Questions:**

* Why not use the labeled data in training the GTA (supervised loss), wouldn't that yield better feature representations? Even if it were implemented as an alternating supervised and unsupervised training procedure for GTA it would have been an interesting experiment.

* The dataset splits in Tables 1 and 2 are very confusing. Why not keep the same splits for both labeled and augmented training sets? In Table 2 what is the proportion of labeled (fine) versus unlabeled (coarse-SBD) data?

* If the Student network is trained only on the labeled data, wouldn't this mean that the Student always sees fewer samples than the other two NNs, how would then the training procedure work? Algorithm 1 does not clarify this part and I suspect you can do this with different batch sizing, but did not find the batch size for any of the models. Maybe clarify this part a bit.

* The authors suggest that training the student with faulty pseudo labels would hinder results. How about training GTA supervised and the Student semi-supervised? Did the authors experiment with this as well, any thoughts on what would the outcome be?

* Do all NNs (GTA, Teacher, Student) have the same type of feature extractor (ResNet-101 pre-trained on ImageNet) and mask predictor (DeepLabv3+)? How about the methods in Tables 1, 2, and 3?

* I do not see a clear advantage of applying the re-weighting mechanism (higher weight on higher confidence) after removing the pixels from the pseudo labels with low confidence. That proposed procedure won't solve the miscalibration or over-confidence phenomenon (detailed L40 - 46). This fact is also confirmed by the results in Table 7 (performance drop after applying re-weighting, which gets corrected after applying Laplace Smoothing). What regions benefit from applying this procedure? Did you see a gain in smaller objects or finer details? A more extensive analysis would have helped.

* In the supplementary material the authors stated that they warm up all the NNs for 1 epoch on labeled data - what are the implications of not doing so? And why just 1 epoch? Just for fair comparisons?

* It would have been interesting to see what is the maximum performance the proposed method could achieve given all the labeled and unlabeled data from any of the benchmarks. I do not see this experiment in any table.

* Suggestion: The "SupOnly" naming is kind of confusing since it's not purely supervised learning right (using only labeled data)? Isn't it the standard Teacher-Student paradigm which is also semi-supervised? Also, stick to one naming convention - Teacher-Student and SupOnly are used interchangeably - Table 4 and Table 5.

**Limitations:**

* The authors have addressed some of their method's limitations (addressing lower performance gains on the augmented sets) but qualitative results and comparisons with other methods and the coarse labels the authors refer to would have maybe a stronger argument - even if they were added as supplementary material.

**Strengths And Weaknesses:**

Strengths:
* Although the proposed extension of the classic teacher-student paradigm is quite simple and straightforward, it is novel and effective
* The paper tackles an interesting topic for the research community
* The method performs considerably better w.r.t previous publications on popular benchmarks
* Extensive experiments with recent relevant related work and ablation studies are also a plus
* Algorithm 1, alongside Figure 1 and 2 complements the text beautifully and further clarify the procedure and enhance the authors' contribution

Weaknesses:
* Some design decisions were not fully explained - e.g. using solely pseudo labels for training GTA
* Some experiments are incomplete - detailed in the "Questions" section
* There are some doubts regarding the experimental setup and reported numbers in the tables (not clear if the methods are reimplementations or original numbers from the papers - this should be specified in the caption of the tables) - since it's stated in the paper that the comparisons are fair.

Others:
* L107 - Typo - "Semi-Supervised Semantic _Segmentation_"
* L124 - Another relevant, even pioneer relevant work for the Teacher-Student framework is [1] which definitely deserves mention.

[1] Croitoru, Ioana, Simion-Vlad Bogolin, and Marius Leordeanu. "Unsupervised learning from video to detect foreground objects in single images." Proceedings of the IEEE International Conference on Computer Vision. 2017.

---

> ### Author Response · Authors · 2022-08-02
> **Response to Reviewer sFdH (Part 1)**
>
> Sincere gratitude to you, especially for the detailed and insightful comments both on our main paper and supplementary materials. We hope our answers can address all of your concerns. We are looking forward to having further discussions with you.
>
> __Q1: Why not use the labeled data in training the GTA (supervised loss), wouldn't that yield better feature representations? Even if it were implemented ...__
>
> Thank you for this insightful comment. In the following table, we try to train GTA with both labeled data and pseudo labels. Following Table 4-7 in our main paper, we take PASCAL VOC 2012, 183 labeled data, with ResNet-101 and DeepLabv3+ as the backbone, to compare the model performance. The student model is still trained with labeled data to update both its encoder and decoder, and its encoder is also updated by GTA via EMA. The teacher model is totally updated by the student model via EMA. We present the comparison results below.
>
>
> | GTA | Student | GTA mIoU | Student mIoU | Teacher ( Final ) mIoU |
> |-------|------|:------:|:------:|:------:|
> | Pseudo Labels | Labeled Data | 70.10 | 72.71 | __73.16__  |
> | Labeled Data + Pseudo Labels | Labeled Data | 71.84 | 71.75 | 72.28 |
>
>
> We can observe that when taking both labeled data and pseudo labels to train GTA, the performance of GTA is improved by 1.74% (70.10% -> 71.84%), but the performance of the student model and teacher model (the final model we take for inference) is decreased by 0.96% (72.71% -> 71.75%) and 0.88% (73.16% -> 72.28%), respectively. It is reasonable that the GTA can learn better representation from more training data (both labeled data and pseudo labels). However, GTA will update the encoder of the student model via EMA, which is also directly learned from labeled data to update both its encoder and decoder. In this approach, the encoder of the student model will be updated by the labeled data with a limited number of images via both EMA and supervised learning, which will possibly cause overfitting and consequently, harm the student model's performance. Meanwhile, since the teacher model is purely updated by the student model via EMA, the performance of the teacher model is also harmed. Therefore, we choose to train GTA with pseudo labels alone. We have updated these discussions in the main paper. (Sec 4.4, Table 7)
>
> __Q2: The dataset splits in Tables 1 and 2 are very confusing ...__
>
>
> These dataset splits strictly follow the previous works for a fair comparison [1][2]. We have revised the notations in Table 1 and 2. We present the number of the fine and coarse labels in dataset split of Table 2 (PASCAL VOC 2012 augmented training set) below. We can observe that this datatset split will first take fine labeled data. Suppose we require 662 or 1323 labeled data, this dataset split will only take the fine ones. When we need more than 1464 labeled data, it will take all of the 1464 fine labeled data, and then get the remaining data from coarse labeled data.
>
>
> | total | fine | coarse|
> |-----|------|-----|
> | 662 | 662 | 0 |
> | 1323 | 1323 | 0 |
> | 2645 | 1464 | 1181|
> | 5291 | 1464 | 3827 |
>
>  [1] Geoff French, Samuli Laine, Timo Aila, Michal Mackiewicz, and Graham Finlayson. Semi-supervised semantic segmentation needs strong, varied perturbations. In British Machine Vision Conference, 2019
>
> [2] Yuliang Zou, Zizhao Zhang, Han Zhang, Chun-Liang Li, Xiao Bian, Jia-Bin Huang, and Tomas Pfister. Pseudoseg: Designing pseudo labels for semantic segmentation. In International Conference on Learning Representations, 2021.
>
> __Q3: If the Student network is trained only on the labeled data, wouldn't this mean that the Student always sees fewer samples than the other two NNs, how would then the training procedure work? Algorithm 1 does not clarify this part ...__
>
>
> In each iteration during training, we sample the same number of labeled and unlabeled data. In other words, the batch size for GTA (using unlabeled data), teacher model (using unlabeled data), and student model (using labeled data) are the same (3 for PASCAL VOC on each GPU, with 4 GPUs; and 4 for Cityscapes on each GPU, with 8 GPUs). During training, the teacher model provides pseudo labels for unlabeled data. GTA learns from these pseudo labels and updates the encoder of the student model via EMA. The student model (both encoder and decoder) further learns from labeled data with supervised training. In this way, the encoder of the student model can gain knowledge from both labeled and unlabeled data. The teacher model is then updated by the student model via EMA. We have revised Algorithm 1 to clarify the procedure.

---

> > ### Author Response · Authors · 2022-08-02
> > **Response to Reviewer sFdH (Part 2)**
> >
> > __Q4: The authors suggest that training the student with faulty pseudo labels would hinder results. How about training GTA supervised and the Student semi-supervised ...__
> >
> >
> > Thank you for providing this important comparison experiment. The results are shown below. We can observe that training GTA supervised and student semi-supervised shows significantly lower performance than our design. It is consistent with our intuition that the student model is not suitable for learning from the noisy pseudo labels directly. We have added these discussions to our revised main paper (Sec 4.4, Table 7).
> >
> >
> > | GTA | Student | mIoU |
> > | :---- | :---- | :----: |
> > | Labeled Data | Pseudo Labels | 66.71 |
> > | Pseudo Labels | Labeled Data | __73.16__ |
> >
> >
> > __Q5: Do all NNs (GTA, Teacher, Student) have the same type of feature extractor ...__
> >
> > All NNs have the same architecture, feature extractor (ResNet-101 pre-trained on ImageNet), and mask predictor (DeepLabv3+). All of the methods in Table 1, 2, 3 take this setting.
> >
> >
> > __Q6: I do not see a clear advantage of applying the re-weighting mechanism ...__
> >
> >
> > Comparison experiment in our original paper is conducted with the situation that there are only 183 labeled data. Under this situation, the lack of labeled data leads to large confidence divergence among pixels, even after confidence filtering. As a result, traditional confidence re-weighting brings about over-penalization. To tackle this phenomenon, we adopt Laplace Smoothing when designing our re-weighting mechanism. The comparison results validate its effectiveness.
> >
> >
> > In addition, we present the ablation study results over different ratios of labeled data below.
> >
> >
> > | Confidence-based Re-weighting | Laplace Smoothing| 92 | 183 | 366 | 732 | 1464 |
> > |---|---|---|---|---|---|---|
> > | x | x | 68.91 | 72.10 | 73.77 | 75.65 | 78.95|
> > | &#10004; | x | 68.04 | 70.67 | 74.33 | 77.31| 79.74|
> > | &#10004; | &#10004; | __70.02__ | __73.16__ | __75.57__ | __78.37__ | __80.47__ |
> >
> >
> > We can observe that when labels are very limited ( e.g. 92 and 183 ), the vanilla confidence-based re-weighting brings about a small performance drop, yet our designed strategy alleviates it by modulating the weight distribution to a smoother one with Laplace Smoothing. On the other hand, when there is more data ( 366, 732, 1464 ), confidence-based re-weighting strategy is beneficial to the model performance, while our designed strategy pushes the performance to a higher level.
> >
> >
> > Besides the quantitative results above, we also add the qualitative comparison results in our revised supplementary material (Appendix A.3). The re-weighting strategy leads to better performance on contour or ambiguous regions.
> >
> >
> > __Q7: In the supplementary material the authors stated that they warm up all the NNs for 1 epoch on labeled data - what are the implications of not doing so? And why just 1 epoch? Just for fair comparisons?__
> >
> >
> > As depicted in Appendix A.1, following previous works, we train all the three models ( GTA, student model, and teacher model ) on labeled data for 1 epoch. After then, GTA and the student model are trained by pseudo labels and labeled data, respectively.
> > Here, we present the results when taking different numbers of epoch here. Our method performs steadily when the warmup epoch varies.
> >
> >
> > | | mIoU |
> > |-----|----|
> > | 0 | 72.93 |
> > | 1 (Reported) | 73.16 |
> > | 2 | 73.58 |
> > | 3 | 73.39|
> >
> >
> > __Q8: It would have been interesting to see what is the maximum performance the proposed method ...__
> >
> >
> > Thank you for providing this interesting experiment. We include all of the labeled and unlabeled data in PASCAL VOC and Cityscapes together to train the model and evaluate the model on PASCAL VOC, with ResNet-101 and DeepLabv3+ as the backbone. The results are shown below (the max column), which demonstrates that our method can tackle external data from other datasets.
> >
> > | | mIoU |
> > |-----|----|
> > | 662 | 77.82 |
> > | 1323 | 80.47 |
> > | 2645 | 80.57 |
> > | 5291| 81.01 |
> > | max | 85.42|
> >
> > __Q9: Suggestion: The "SupOnly" naming is kind of confusing since it's not purely supervised learning right (using only labeled data)? ...__
> >
> >
> > SupOnly actually means using merely the labeled data to train the model. Teacher-Student means the standard teacher-student framework, which takes both labeled and unlabeled data during training. We have clarified it in revision (Table 1 in the main paper).
> >
> > __Q10: Typos and missed related works.__
> >
> > Thanks. We have fixed these typos and added the related work in our revised paper.

---

### Official Review · Reviewer_3UiM · 2022-07-12

**Rating:** 6
**Confidence:** 3
**Soundness:** 3 good
**Presentation:** 2 fair
**Contribution:** 3 good

**Summary:**

The authors argue that the unlabeled data with pseudo labels is more beneficial to the feature extractor than the mask predictor in a semantic segmentation framework. Based on this, they introduce the Gentle Teaching Assistant model which learns from pseudo labels and (only) transfers the feature representation to the student model.



**Questions:**

* Why not utilize both labeled data and pseudo labels to train GTA? It does not make sense to me that fewer data can learn better feature representation.
* How to initialize GTA, is it initialized randomly and trained from scratch?
* Is it necessary to maintain the similarlity between the parameters of GTA and the student model? If the two models are very different from each other, does it make sense to transfer the knowledge through EMA?
* More qualitative results (maybe in the supp) can help improve the paper. Currently, there is only one figure of visualization result in the supplementary and main text.
* Missing ablation study of EMA hyper-parameters.
* In Table 5, is the difference between ours and the original EMA that only the decoder parameters are updated? If so, maybe it’s better to express it as EMA (all parameters) and EMA (decoder).

Typos:
* line 109, is y^u a typo?
* eq1: there is no k in p^u_{i, j}, although we can guess that k is kth term in p^u_{i, j},
* eq3: why the CE loss is defined on data and pseudo labels rather than prediction and labels?
* table 1: does 10582 should be 1464?



**Limitations:**

The authors point out one limitation that the proposed method performs better on high-quality labeled data.

**Strengths And Weaknesses:**

### Strengths:
+ A novel gentle teaching assistant model that utilizes pseudo labels to aid feature representation learning.
+ The paper is well motivated and the logic is easy to follow.
+ Extensive ablation studies are provided.


### Weaknesses:
- Considering that GTA only uses pseudo labels, it is unclear to me why GTA is able to transfer better feature representation to the student model.
- The presentation could be improved. The method section could be more concise (see the questions).
- Not enough qualitative results (see the questions).

Post rebuttal:
I'd keep the current rating based on the rebuttal. Although the authors provided reasonable answers to the Q1 (similar concerns also appears in Reviewer sFdH), the implementation is a bit heuristic and need further discussions in the revised paper.

---

> ### Author Response · Authors · 2022-08-02
> **Response to Reviewer 3UiM**
>
> Sincere gratitude to you, especially for the constructive comments on our method design. We hope our answers can address all of your concerns. We are looking forward to having further discussions with you.
>
> __Q1: Why not utilize both labeled data and pseudo labels to train GTA? It does not make sense to me that fewer data can learn better feature representation.__
>
>
> Thank you for this insightful comment. In the following table, we try to train GTA with both labeled data and pseudo labels. Following Table 4-7 in our main paper, we take PASCAL VOC 2012, 183 labeled data, with ResNet-101 and DeepLabv3+ as the backbone, to compare the model performance. The student model is still trained with labeled data to update both its encoder and decoder, and its encoder is also updated by GTA via EMA. The teacher model is totally updated by the student model via EMA. We present the comparison results below.
>
>
> | GTA | Student | GTA mIoU | Student mIoU | Teacher ( Final ) mIoU |
> |-------|------|:------:|:------:|:------:|
> | Pseudo Labels | Labeled Data | 70.10 | 72.71 | __73.16__  |
> | Labeled Data + Pseudo Labels | Labeled Data | 71.84 | 71.75 | 72.28 |
>
>
> We can observe that when taking both labeled data and pseudo labels to train GTA, the performance of GTA is improved by 1.74% (70.10% -> 71.84%), but the performance of the student model and teacher model (the final model we take for inference) is decreased by 0.96% (72.71% -> 71.75%) and 0.88% (73.16% -> 72.28%), respectively. It is reasonable that the GTA can learn better representation from more training data (both labeled data and pseudo labels). However, GTA will update the encoder of the student model via EMA, which is also directly learned from labeled data to update both its encoder and decoder. In this approach, the encoder of the student model will be updated by the labeled data with a limited number of images via both EMA and supervised learning, which will possibly cause overfitting and consequently, harm the student model's performance. Meanwhile, since the teacher model is purely updated by the student model via EMA, the performance of the teacher model is also harmed. Therefore, we choose to train GTA with pseudo labels alone. We have updated these discussions in the main paper. (Sec 4.4, Table 7)
>
> __Q2: The presentation could be improved. The method section could be more concise.__
>
> We have revised the method section in our revised paper to make it more concise and clear.
>
>
> __Q3: How to initialize GTA ...__
>
>
> As depicted in Appendix A.1, following previous work, we train all the three models ( GTA, student model, and teacher model ) on labeled data for 1 epoch, which works as a warmup and makes the training process more stable. After then, GTA and the student model are trained by pseudo labels and labeled data, respectively.
>
>
> __Q4: Is it necessary to maintain the similarity between the parameters of GTA and the student model? If the two models are ...__
>
>
> Thank you for this insightful comment. We try to maintain the similarity between GTA and the student model via L2 penalty on distances of parameters. Following Table 4-7 in our main paper, we evaluate the models with 183 labeled data on PASCAL VOC 2012, with ResNet-101 and DeepLabv3+ as the backbone.
>
>
> | | GTA | Student | Teacher|
> |-------|------|------|------|
> | Ours + Similarity | 69.55 | 71.86 | 72.47 |
> | Ours | 70.10 | 72.71 | __73.16__ |
>
>
> The results show that maintaining the similarity between the parameters of GTA and the student model is incapable of boosting model performance. And we conjecture that since the GTA performs worse than the student model, closing the distances of all their parameters even slightly harms the performance of the student model.
>
>
> __Q5: More qualitative results (maybe in the supp) can help improve the paper.__
>
>
> Thanks for your suggestions. We have added more qualitative results in our revised supplementary material (Appendix A.3).
>
>
> __Q6: Missing ablation study of EMA hyper-parameters.__
>
>
> We present the ablation study of EMA hyper-parameters here and have added it to our revision (Table 9 in Appendix). Our method performs stably over different EMA hyper-parameters.
>
>
> | | mIoU |
> |-----|------|
> | 0.98 | 72.65 |
> | 0.99 (Ours) | 73.16 |
> | 0.999 | 73.44 |
> | 0.9999 | 73.57|
>
>
> __Q7: In Table 5, is the difference between ours and the original EMA that ...__
>
>
> The encoder parameters of the student model are updated by GTA via EMA, and its decoder is trained by supervised learning. We have changed our expression in revision (Table 5 in the main paper).
>
>
> __Q8: Typos.__
>
>
> Thanks. We have fixed these typos in our revised paper.

---

### Author Response · Authors · 2022-08-02
**Response to all our reviewers.**

We thank all our reviewers for your distinguished efforts and insightful comments. We have answered your questions in our responses, hoping that we can address your concerns.


We have also uploaded the revised paper and supplementary materials (the modifications are present in blue color). Here, we summarize our main modifications.


__1. Method design analysis:__ We add the analysis of our model design in Sec 4.4 in our revised paper, where we compare other different method designs with our proposed one and demonstrate that our method shows the strongest performance. We also try to modify Sec 3 to make it more concise and clear.

__2. Clarification of experiment settings:__ We have modified notations in Table 1-3 to avoid them from being confusing.

__3. Limitation analysis:__ The analysis of limitations in our original paper may cause some misunderstandings. We have analyzed the limitation from other perspectives.

__4. Clarification of implementation details:__ We have added more implementation details of our method and experiment in our revised Appendix A.1, including hyper-parameters and random seeds. We also conduct a hyper-parameter sensitivity analysis in Appendix A.2.3.

__5. More visualization results:__ We present more visualization results in revised Appendix A.3.

__6. Typos and missed related works:__ We have fixed the typos and added the works our reviewers point out.


Finally, sincere gratitude to our reviewers. We are looking forward to having further discussions with you.

---

### Meta-Review · Area_Chair_vhVu · 2022-08-27

**Recommendation:** Accept
**Confidence:** Certain

**Metareview:**

The paper was reviewed by four expert reviewers in the field. The initial ratings were three weak accept and one weak reject.

In the response to reviewer sFdH (who gave Weak reject), the authors clarify all the questions from the reviewer, including using labeled data in GTA, the data split, training details, advantages of re-weighting mechanism. While the reviewer sFdH did not acknowledge the rebuttal, the AC believes that these questions have been sufficiently addressed.

Given the novel approach, extensive quantitative evaluation, and clear writing, the AC agrees with the three reviewers and recommends to accept.

**Award:**

No

---

### Decision · Program_Chairs · 2022-09-14

Accept